# The intricate relationship of G-Quadruplexes and bacterial pathogenicity islands

Bo Lyu*, Qisheng Song*

Division of Plant Science and Technology, University of Missouri, Columbia, United States

**Abstract** The dynamic interplay between guanine-quadruplex (G4) structures and pathogenicity islands (PAIs) represents a captivating area of research with implications for understanding the molecular mechanisms underlying pathogenicity. This study conducted a comprehensive analysis of a large-scale dataset from reported 89 pathogenic strains of bacteria to investigate the potential interactions between G4 structures and PAIs. G4 structures exhibited an uneven and non-random distribution within the PAIs and were consistently conserved within the same pathogenic strains. Additionally, this investigation identified positive correlations between the number and frequency of G4 structures and the GC content across different genomic features, including the genome, promoters, genes, tRNA, and rRNA regions, indicating a potential relationship between G4 structures and the GC-associated regions of the genome. The observed differences in GC content between PAIs and the core genome further highlight the unique nature of PAIs and underlying factors, such as DNA topology. High-confidence G4 structures within regulatory regions of *Escherichia coli* were identified, modulating the efficiency or specificity of DNA integration events within PAIs. Collectively, these findings pave the way for future research to unravel the intricate molecular mechanisms and functional implications of G4-PAI interactions, thereby advancing our understanding of bacterial pathogenicity and the role of G4 structures in pathogenic diseases.

## eLife assessment

This **fundamental** study explores the relationship between guanine-quadruplex structures and pathogenicity islands in 89 bacterial strains representing a range of pathogens. Guanine-quadruplex structures were found to be non-randomly distributed within pathogenicity islands and conserved within the same strains. These **compelling** findings shed light on the molecular mechanisms of Guanine-quadruplex structure-pathogenicity island interactions and will be of interest to all microbiologists.

## Introduction

The discovery of the DNA double helix by Watson and Crick in 1953 revolutionized our understanding of genetics and laid the foundation for the modern field of molecular biology (*Watson and Crick, 1953*). Nonetheless, the intricate nature of DNA continues to surprise us even today. One such captivating feature is the DNA guanine (G)-quadruplex (G4) structure, a unique arrangement that defies the conventional double helix (*Rhodes and Lipps, 2015*; *Spiegel et al., 2020*). A G4 consists of four guanine bases and is stabilized by Hoogsteen hydrogen bonds. These stacked tetrads are interconnected by loop regions, which can vary in length and sequence, adding further complexity to the structure (*Figure 1*). It is important to consider the inherent directionality of nucleic acids, with all four

*For correspondence:
bl3pt@missouri.edu (BL);
SongQ@missouri.edu (QS)

**Competing interest:** The authors declare that no competing interests exist.

strands having the possibility to run in the same 5' to 3' direction, referred to as 'parallel,' or alternatively, they can run in different directions, known as 'antiparallel.'. G4 regions can be very stable in vitro, particularly in the presence of K$^+$ (*Stegle et al., 2009*). G4 structures are often found in regions of the genome with crucial regulatory functions, such as telomeres, promoters, and enhancers (*Rhodes and Lipps, 2015*; *Huppert, 2010*). These structures play a role in various biological processes, including gene expression, DNA replication, and telomere maintenance (*Rhodes and Lipps, 2015*; *Zybailov et al., 2013*). Further research into G4 structures will undoubtedly uncover new insights into their functions and facilitate the development of innovative technologies.

PAIs are genomic regions that contribute to the virulence and pathogenic potential of various microorganisms (*Schmidt and Hensel, 2004*; *Groisman and Ochman, 1996*). PAIs are distinct segments of the bacterial genome that exhibit unique characteristics compared to the rest of the DNA (*Hacker and Kaper, 2000*). They are often large in size, ranging from tens of kilobases to hundreds of kilobases, and can be integrated into the chromosome or exist as extra-chromosomal elements, such as plasmids. PAIs often exhibit close proximity to tRNA genes, suggesting a putative mechanism where tRNA genes act as anchor points for the integration of foreign DNA acquired through horizontal gene transfer (*Figure 1E*). One notable feature is their variable GC content, which tends to deviate from the average GC content of the genome in various organisms, such as *Streptomyces* (*Kers et al., 2005*), *Salmonella* (*Kombade and Kaur, 2021*), and *Yersinia* (*Carniel, 1999*). PAIs typically contain clusters of genes involved in pathogenesis, including those encoding secretion systems (e.g. LEE (locus of enterocyte effacement) in *Escherichia coli*), superantigen (e.g. SaPI1 and SaPI2 in *Staphylococcus aureus*), and enterotoxin (e.g. *she* PAI in *Shigella flexneri*). PAIs can be acquired through the transfer of mobile genetic elements, such as plasmids, phages, or integrative and conjugative elements (ICEs), facilitating the incorporation of pathogenicity-associated genes into the recipient genome (*Schmidt and Hensel, 2004*; *Syvanen, 2012*; *Chen et al., 2015*). One question raised in PAI is that PAIs often exhibit distinct base composition (G+C contents) compared to the core genome. The underlying reasons for this variation remain unknown, but the preservation of a genus- or species-specific base composition represents a noteworthy characteristic of bacteria (*Schmidt and Hensel, 2004*). Schmidt and Hensel proposed a hypothetical mechanism to explain the observed variation, suggesting that factors such as DNA topology and codon message in the virulence regions present could contribute to the preservation of the distinct base composition (*Schmidt and Hensel, 2004*). Hopefully, the availability of genome sequences from pathogenic bacteria and their non-pathogenic counterparts presents an exceptional opportunity to explore the intricate structure variance and underlying mechanisms within PAIs.

Growing evidence has shown that G4 structures exhibit a striking colocalization with functional regions of the genome, and their high conservation across different species suggests a selective pressure to maintain these sequences at specific genomic regions (e.g. genome islands, resistance islands, CpG islands, and PAIs) (*Rhodes and Lipps, 2015*; *Frees et al., 2014*; *König et al., 2010*). The possibility of interactions between G4 structures and pathogens has been suggested, although this field of study is still in its nascent phase. Some studies observed that bacterial genomes possess G4-forming sequences within their genome regions (*Yadav et al., 2021*; *Harris and Merrick, 2015*). G4 structures are formed by G-rich DNA sequences, and their stability is influenced by the G+C content and arrangement of G tetrads. Interestingly, PAIs often exhibit an altered GC content, putatively contributing to the propensity of G4 structure formation within these regions. The G4 structures in PAIs might modulate the accessibility of transcription factors, DNA-binding proteins, or RNA polymerase in pathogens, as documented in eukaryotes (*Rhodes and Lipps, 2015*; *Varshney et al., 2020*), thereby influencing the expression of virulence-associated genes (*Cahoon and Seifert, 2009*). The formation of G4 structures within PAIs may serve as an additional layer of regulation that fine-tunes the expression of genes critical for pathogenesis. Hence, the investigation of G4 structures within PAIs may open new avenues for the development of therapeutic strategies aimed at disrupting the regulatory mechanisms of pathogenicity-associated genes.

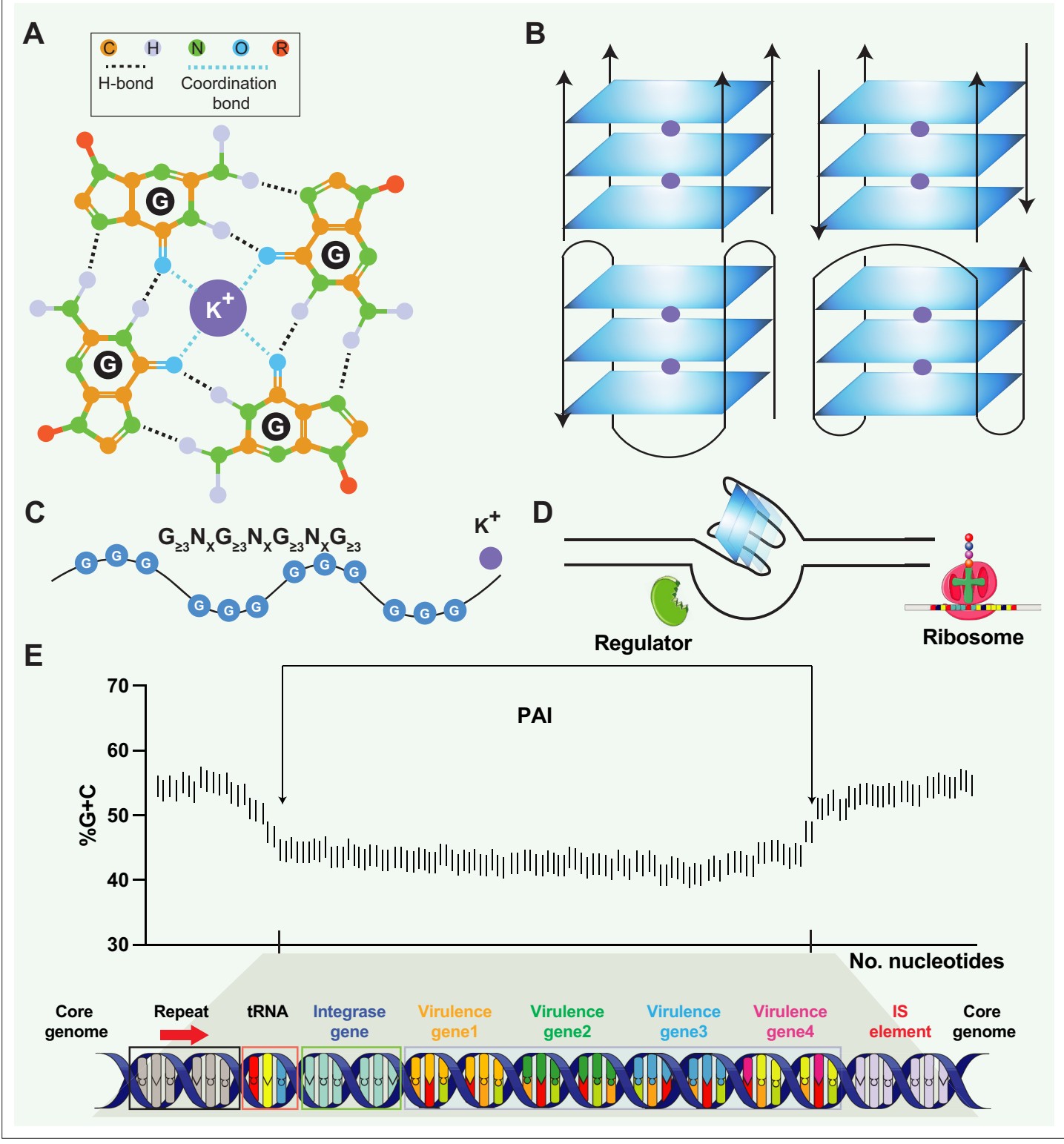

**Figure 1.** Structural and functional aspects of G-quadruplex (G4) structures and pathogenicity islands (PAIs). (**A**) Schematic representation of a guanine tetrad stabilized by Hoogsten base pairing and a positively charged central ion, illustrating the key elements of G4 structures. (**B**) Structural heterogeneity of G4 structures. G4 structures exhibit polymorphism and can be categorized into different families, such as parallel or antiparallel, based on the orientation of the DNA strands. They can fold either intramolecularly or intermolecularly, leading to diverse structural configurations. (**C**) General sequence formula for G4, highlighting the repeated occurrence of guanine-rich sequences that form G4 structures. (**D**) Regulatory roles of G4 in transcription. G4 can regulate transcription by blocking RNA polymerase from binding to promoter sequences or aiding in single-stranded

*Figure 1 continued on next page*

*Figure 1 continued*

DNA (ssDNA) formation, thereby enhancing transcription. (**E**) General structure of pathogenicity islands (PAI). PAIs are characteristic regions of DNA found within the genomes of pathogenic bacteria, distinguishing them from nonpathogenic strains of the same or related species. Repeat sequences are DNA segments duplicated within the PAI and can serve as recognition sites for various enzymes involved in the integration and excision of the PAI from the bacterial chromosome. tRNA genes act as anchor points for the insertion of foreign DNA acquired through horizontal gene transfer. Virulence genes encode proteins or factors that play crucial roles in the virulence and pathogenicity of the bacterium, contributing to adhesion, invasion, immune evasion, toxin production, or other pathogenic mechanisms. Insertion elements include transposons, bacteriophages, or plasmids, enabling the PAI to be transferred between bacterial cells and potentially disseminated to different strains or species.

## Results

### Genomic information, PAI patterns, and the presence of G4 structures in 89 reported pathogenic strains

A dataset of PAIs was compiled from 89 reported pathogenic strains of bacteria, encompassing 222 distinct types of PAIs. Pathogens exhibiting similar PAIs displayed closely clustered patterns on phylogenetic branches, such as LEE in *E. coli* strains (*Figure 2A*). Additional information, including the genome length (bp), G+C content (%), rRNA density, tRNA density, and PAI length (bp), was

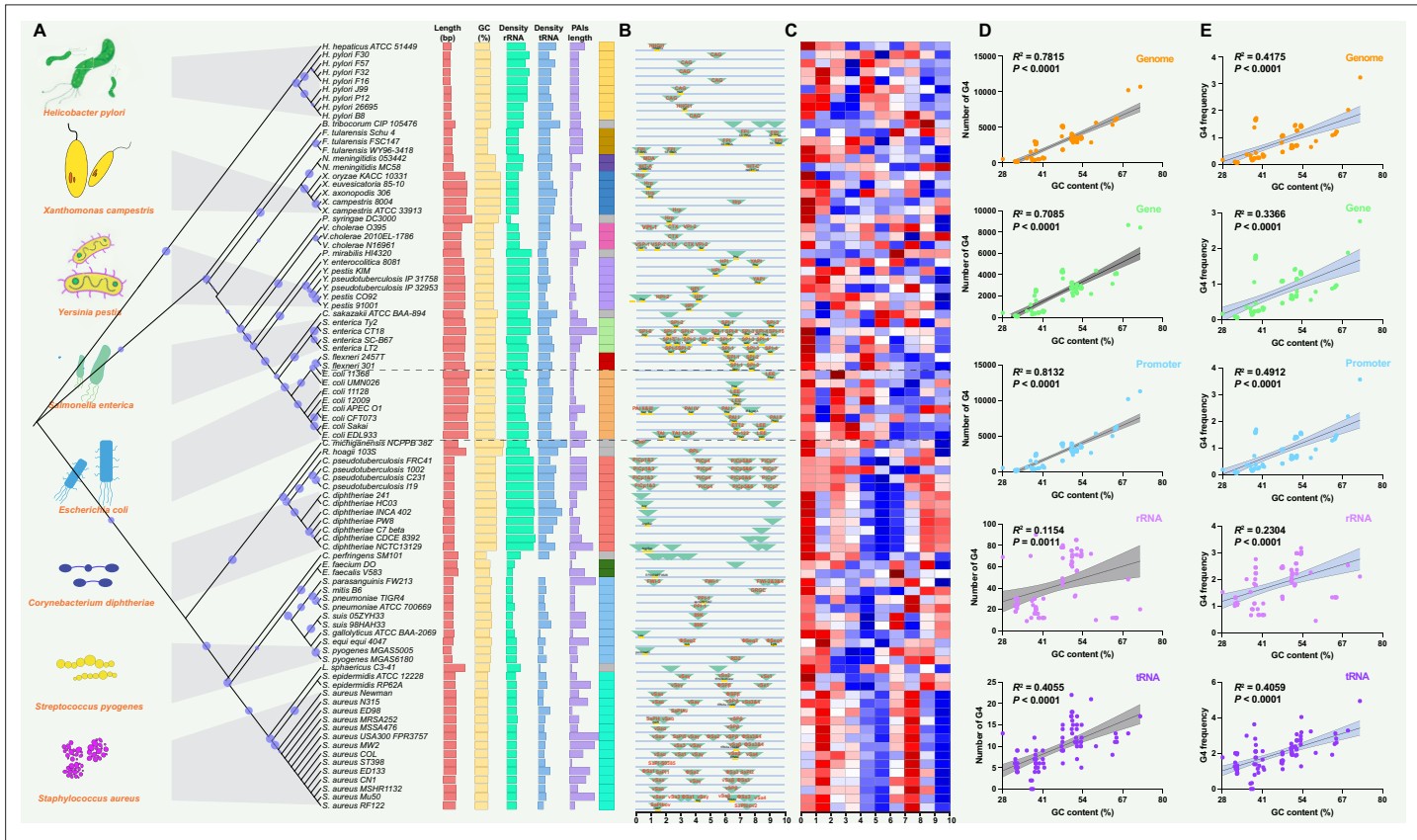

**Figure 2.** Analysis of pathogenicity islands (PAIs) and G-quadruplexes (G4) in pathogen genomes. (**A**) Phylogenetic analysis of pathogen genomes based on 89 bacterial strains, showing the evolutionary relationships among species. Additional genomic information, including genome size, GC content, rRNA density, tRNA density, and PAI length, is provided. The same color indicates the same species. (**B**) Genomic location of specific PAIs in bacterial genomes, divided into ten regions. PAIs are represented by green triangles, and their names are indicated. The tRNA insertion sites are also marked. (**C**) Heatmap illustrating the relative abundance of G4 structures in bacterial genomes, divided into ten regions. Red indicates a higher relative abundance, while blue indicates a lower relative abundance. (**D** & **E**) Correlation analysis between the number of G4 structures, the frequency of G4 structures, and GC content in various genomic features, including the whole genome, genes, promoters, rRNA, and tRNA. R-squared and p-values were derived through linear regression analysis performed in GraphPad Prism.

The online version of this article includes the following figure supplement(s) for figure 2:

**Figure supplement 1.** Correlation analysis between the number of G4s, the frequency of G4s, and GC content in various genomic features, including the whole genome, genes, promoters, rRNA, and tRNA, using G4 scores of 1.4 and 1.6.

present and showed conserved patterns in the same species (*Figure 2A*; *Supplementary file 1a*). PAIs commonly exhibit mosaic-like patterns, exemplified by the presence of distinct PAIs like FPI in *Francisella tularensis*, SaPIbov in *Staphylococcus aureus*, and Hrp PAI in *Xanthomonas campestris* (*Figure 2B*). Many PAIs were present associated with tRNAs, such as the insertions of tRNA$^{Thr}$, tRNA$^{Phe}$, and tRNA$^{Gly}$ in *E. coli* strains (*Figure 2B*; *Supplementary file 1b*). The presence of PAIs distributes in similar genomic regions across different pathogens or strains, showing non-random patterns and functionally clustered. Employing the G4Hunter search algorithm, the study identified a total of 225,376 putative G4 sequences in these 89 pathogenic genomes (*Supplementary file 1a*). The heatmap also showed that the number of G4 structures was diverse in the pathogen genomes (*Figure 2C*).

## Interaction between PAIs and G4 structures in different genomic features

The analysis of G4 structures across all pathogen species demonstrated a positive correlation between the number of G4 structures and the GC content in various genomic features, including the whole genome, gene, promoter, rRNA, and tRNA regions (*Figure 2D*). The frequency of G4 structures, measured as the frequency of predicted G4-forming sequences per 1000 base pairs (bp), also showed a positive correlation with the GC content across the analyzed genomic elements (*Figure 2E*). A G4 score of 1.4 and 1.6 consistently supported a positive correlation between the number and frequency of G4 structures and the GC content across diverse genomic features (*Figure 2—figure supplement 1*). Additionally, this study observed that the GC contents in the genome region were significantly higher compared to the corresponding PAIs region that was classified into five parts according to the genome datasets (*Figure 3A–E*). Nonetheless, this study noted a unique pattern in the frequency of G4 structures within diverse regions of the PAIs, particularly in regions with GC contents less than 30% and greater than 60%.

## Putative functions of G4 structures in PAIs

The study used *E. coli* as an example to investigate the potential regulatory role and function of genes covered by G4 structures in PAIs. *E. coli* contains at least ten types of PAIs in different strains, and one of the well-known PAIs is LEE (*Figure 3F*), harboring genes responsible for causing attaching and effacing lesions (*Franzin and Sircili, 2015*; *Jores et al., 2004*). One stable G4 structure with a G4Hunter score of 1.6 was identified at position 37,085 in the LEE PAI of *E. coli* str. O103:H2 12009 (*Figure 3G*), located between an IS element and a tRNA insertion site. The tRNA region generally contains a higher G4 frequency compared with transfer-messenger RNA (tmRNA) and rRNA regions in the bacterial genome (*Bartas et al., 2019*). Interestingly, this G4 structure was found in *E. coli* str. O103:H2 12009 was present in close proximity to a tRNA region, suggesting a potential regulatory role of G4 structures in the tRNA gene, or upstream- and downstream-genes that are responsible for LEE virulence. Additionally, another stable G4 sequence with a score of 1.381 was discovered at position 12,457 in the *E. coli* str. CFT073 PAI II to provide more evidence of G4 in PAI regions (*Figure 3H*). Functional enrichment analysis was conducted to explore the putative functions of G4-covered genes in the two *E. coli* strains (*Supplementary file 1c and d*). The results revealed that the genes covered by G4 structures were predominantly involved in genetic information processes, including DNA binding, DNA integration, and nucleic acid metabolism processes (*Figure 3I & J*).

## Discussion

This study found that the non-random distribution of G4 structures within PAIs across different bacterial species, signifies a potential regulatory role in bacterial pathogenicity. The conservation of G4 structures within the same pathogenic strains suggests a crucial and possibly conserved function in regulating pathogenic traits. The findings are similar to previous reports that showed that the G4 structures display uneven distribution patterns in eukaryotic and prokaryotic genomes and are conserved evolutionary groups (*Bartas et al., 2019*; *Du et al., 2009*; *Puig Lombardi et al., 2019*). To understand the origin of G4 structures within PAIs, we hypothesized that these G4 sequences could be acquired through three types of horizontal gene transfer mechanisms: conjugation, transformation, and transduction (*Figure 3K*). These mechanisms serve as means for genetic material exchange between different organisms. Considering the presence of G4 sequences within the PAIs,

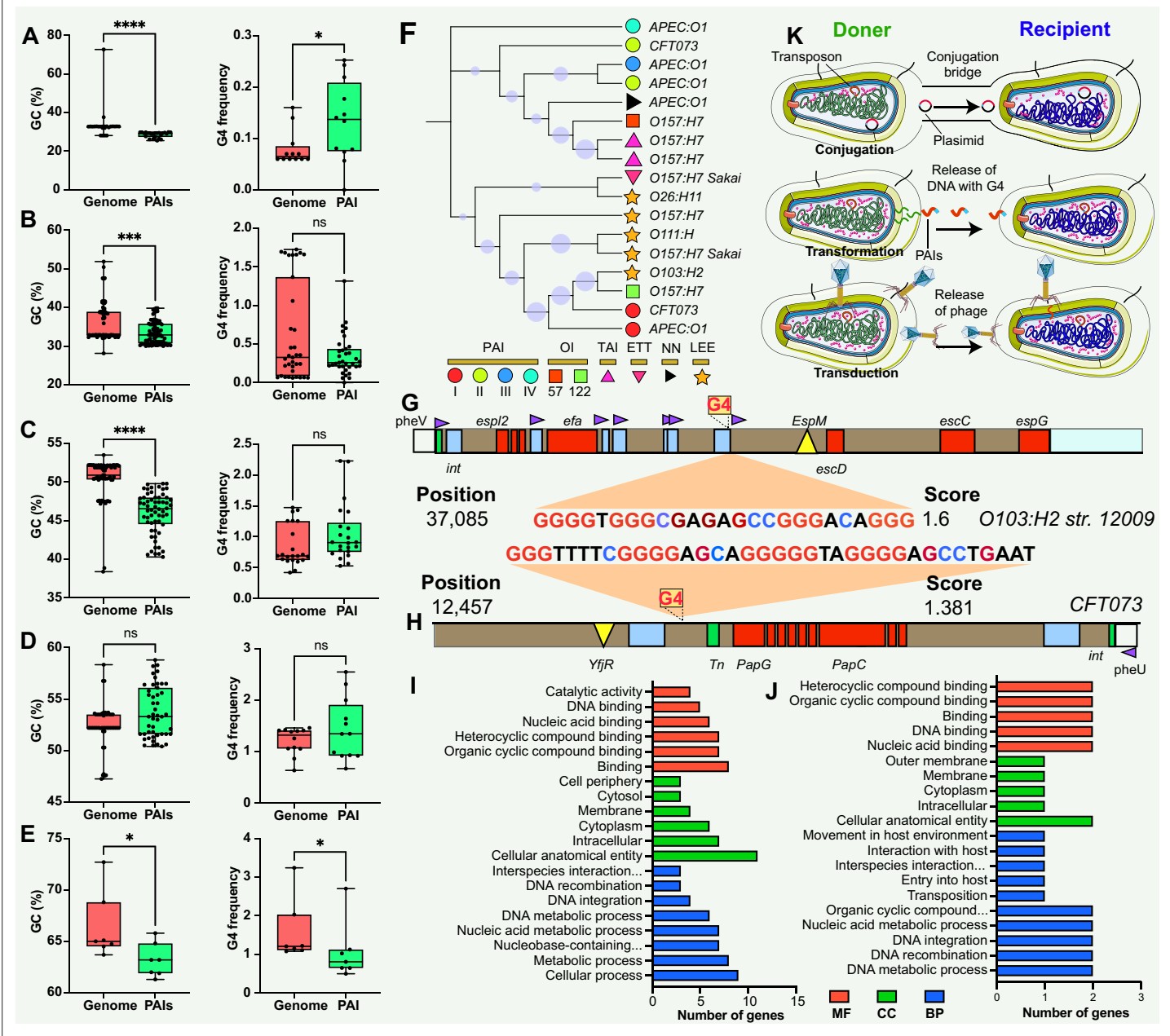

**Figure 3.** Comparison and functional annotation of G-quadruplexes (G4) within pathogenicity islands (PAIs). (A–E) Comparison of GC content (left panel) and GC frequency (right panel) between the genome and PAIs, categorized into five regions (20–30%, 30–40%, 40–50%, 50–60%, and 60–70%). */**/***/**** indicates significant difference (p<0.05/0.01/0.001/0.0001). (F) Evolutionary relatedness of 10 types of PAIs (categorized into six main categories) in *E. coli* strains. (G & H) Examples of G4 structures within PAIs in *E. coli* strains. The gray bar represents the virulence region, the red box indicates a virulence gene, the blue box represents an insertion site region or repeat, the green box denotes an integrase, the purple triangle indicates a tRNA insertion site, and the yellow triangle indicates an effector. (I &J) Functional annotation analysis of G4-covered genes within PAIs in two *E. coli* strains, including biological process (BP), cellular component (CC), and molecular function (MF) categories. (K) Hypotheses on the origin of G4 structures within PAIs, involving gene horizontal transfer mechanisms (conjugation, transduction, and transformation).

it is plausible that these sequences are transferred along with the PAIs through these horizontal gene transfer mechanisms. Additionally, the presence of G4 structures within the promoter, rRNA, and tRNA regions may have functional implications for the regulation of DNA replication, ribosome biogenesis, protein synthesis, and other RNA-related processes (*Zybailov et al., 2013*; *Ivanov et al., 2014*; *Mestre-Fos et al., 2019*). Throughout evolution, there seems to be a greater frequency of G4

structures in regulatory genes, such as the tRNA region, compared to other genes, enabling intricate control of gene expression in signal transduction pathways (*Wu et al., 2021*).

The study found that the genomic regions surrounding the PAIs (i.e. core genome) tend to have a higher GC content than PAI regions, which was consistent with the fact that PAIs often exhibit distinct base compositions compared with the core genome (*Schmidt and Hensel, 2004*). The variation was explained by the presence of G4 sequences within the PAIs, whereas the results were surprising. This study observed a distinct pattern in the frequency of G4 structures within different regions of the PAIs. This differential distribution of G4 structures suggests that (i) specific genomic segments within the PAIs may be more prone to induce G4 formation discrepancy; (ii) the variation of base composition between core genome and PAIs is partially correlated with the presence of G4 structures; (iii) the frequency of G4 structures in PAIs present stable as the core genome in the most situation; (iv) an alternative hypothesis, other factors, such as i-motif (i.e. the anti-G4 structure) and CpG island, may work synergistically with G4 and potentially contribute the base composition variation (*Deaton and Bird, 2011*; *Sushmita, 2020*).

Enrichment analysis indicated a predominant involvement of these G4-covered genes in genetic information processes, encompassing DNA binding, DNA integration, and nucleic acid metabolism. This suggests that G4 structures may play a regulatory role in these essential cellular processes, especially gene expression and DNA-related functions. For instance, G4 structures in the promoter regions of certain transcription factors may influence their binding affinity to DNA and subsequently affect downstream gene expression patterns (*Niu et al., 2018*; *Xiang et al., 2022*). These elements frequently utilize DNA integration mechanisms mediated by integrases, recombinases, or transposases to transfer or incorporate genetic material into the bacterial genome (*Arkhipova and Rice, 2016*; *Wozniak and Waldor, 2010*). One compelling illustration is a study that identified a 16-base pair cis-acting G4 sequence near the pilin locus in *Neisseria gonorrhoeae*, demonstrating its pivotal role in antigenic variation and directing recombination to a specific chromosomal locus (*Cahoon and Seifert, 2009*; *Cahoon and Seifert, 2013*). Disruption of the G4 structure in this context impeded pilin antigenic variation and recombination, highlighting its significance in immune evasion mechanisms. Additionally, considering the distance between G4 structures and the beginning site of gene (e.g. transcription start site (TSS)) in the analysis of promoter regions is pivotal for a comprehensive understanding of their regulatory impact on gene expression (*Huppert, 2010*). The spatial proximity to the TSS influences interactions with regulatory elements, potentially modulating the binding of transcription factors and RNA polymerase. This spatial relationship affects accessibility, with G4 structures closer to the TSS potentially acting as direct impediments to transcription initiation. Acknowledging these spatial nuances would provide crucial insights into the functional implications of G4 structures in promoters.

Overall, the conserved evolutionary relatedness of PAIs, the detection of stable G4 structures in specific genomic positions, and the enrichment of G4-covered genes in genetic information processes collectively support the hypothesis that G4 structures may have regulatory functions in key biological processes in pathogens. However, it is important to acknowledge and address certain limitations that could potentially affect the interpretation of the results. One such limitation is the reliance on genome sequences obtained from external laboratories and datasets, which introduces a level of uncertainty regarding the accuracy and completeness. Furthermore, the dynamic nature of bacterial genomes, including genetic rearrangements and horizontal gene transfer events, can complicate the accurate assembly and annotation of genome sequences. Lastly, the stability of G4 structures seems to be important for their function according to recent evidence (*Jara-Espejo and Line, 2020*). Hence, exploring the relationship between G4 stability and function is a valuable and intriguing topic that could provide insights into the nuanced ways G4 structures contribute to cellular processes and potentially offer new avenues for therapeutic interventions or molecular engineering. To overcome these constraints, fostering collaboration among research teams and participating in data-sharing endeavors becomes imperative to guarantee access to high-quality genome data for exhaustive analyses. Moreover, it is crucial to interpret the results with caution and continue refining this understanding through validation experiments and collaborative efforts.

## Methods
### Selection and extraction of DNA sequences
A total of 89 genomes corresponding to the identified pathogens from the Pathogenicity Island Database (PAIDB) were included in the study. The complete bacterial genomic DNA sequences and their

corresponding annotation files in.gff and.fna formats were obtained from the Genome database of the National Center for Biotechnology Information (NCBI, https://www.ncbi.nlm.nih.gov/genome). To ensure the reliability and completeness of the dataset, only completely assembled genomes were included in the analysis. To avoid redundancy and incomplete sequences, one representative genome was selected for each species or strain. The selection of representative genomes was based on a careful examination of the supplementary material (*Supplementary file 1a*) accompanying the study. TBtools II (Toolbox for Biologists, RRID:SCR_023018, v2.042) (https://cj-chen.github.io/tbtools), a versatile bioinformatics tool with extensive applications in both eukaryotes and prokaryotes (*Chen et al., 2020*; *Chen et al., 2023*), was employed for extracting genomic sequences. This tool facilitated the retrieval of gene regions, promoters (2 kb upstream of the genes), tRNA regions, and rRNA regions from the selected genomes. PAI regions were downloaded following previously documented information in PAIDB (*Supplementary file 1a and b*). Default thresholds and parameters were applied during extraction to maintain consistency across all genomes.

## Data process and detection of G4 structures in genomic features

The G4Hunter algorithm, a widely used tool for G4 prediction, was employed for the identification of G4 motifs in the genomic sequences (*Brázda et al., 2019*). The G4Hunter parameters were set to a window size of '25' and a G4 score threshold of 1.2, which ensured the identification of potential G4 sequences (*Bartas et al., 2019*; *Brázda et al., 2020*). The study additionally utilized G4 scores of 1.4 and 1.6 as a means of cross-verification for the results. The study quantified the predicted number of putative G4-forming sequences within different genomic features, including the whole genome, gene, promoter, tRNA, rRNA, and PAI regions. The density of G4 motifs was determined by dividing the number of G4 sequences by the total length of the genome, while the length ratio of G4 motifs was calculated by dividing the total length of the G4 sequences by the total length of the genome.

## Relationship between G4 structures and PAIs

The heatmap was used to show the distribution of G4 motifs in the genome divided by ten parts as PAI regions using R package 'pheatmap.' The correlation between the number of G4 structures and the GC content was analyzed across various genomic elements, including the whole genome, gene, promoter, rRNA, and tRNA regions. The analysis utilized the *R*-squared value ($R^2$) to determine the fit goodness of the correlation. The correlation's significance was evaluated through p-values along with a 95% confidence interval. Subsequently, a ROC analysis, yielding an area greater than 0.90, was employed to quantify sensitivity and specificity. The GC content in the genome regions and corresponding PAI regions was compared and classified into different ranges to explore the variation in base composition. GraphPad Prism (V.5.02, GraphPad Software, Inc) was employed to conduct Normality and Lognormality Tests. The K-S test and F-test were used to assess normal distribution and variances, and the Student's t-test was used to identify significant differences.

## Phylogenetic tree construction

The exact Taxonomy ID (taxid) for each analyzed group was obtained from the NCBI Taxonomy Database using the Taxonomy Browser. The Neighbor-Joining (NJ) method was employed to construct the phylogenetic trees for the analyzed groups. The phylogenetic trees were generated using MEGA11 software (https://www.megasoftware.net/), which offers robust algorithms and comprehensive tools for phylogenetic analysis. To assess the reliability and statistical support of the phylogenetic tree branches, bootstrap analysis was performed. One thousand bootstrap replicates were used to estimate the confidence levels of the branching patterns in the phylogenetic trees. The phylogenetic trees, along with the bootstrap support values, were displayed and visualized using the Interactive Tree of Life (ITOL) platform (https://itol.embl.de/).

## Gene functional annotation

The gene sequences covered by G4 structures within PAIs were subjected to gene ontology (GO) annotation (https://geneontology.org/). The gene sequences were translated into protein sequences using the Expasy online toolkit (https://web.expasy.org/translate/). This tool performs the translation based on the standard genetic code, converting the DNA nucleotide sequence into its corresponding amino acid sequence. The GO annotation database assigned GO terms to the protein sequences

based on their predicted functions and known biological process (BP), molecular function (MF), and cellular component (CC). Fisher's exact test was employed to determine the statistical significance of the enrichment results. The obtained *p*-values indicated the overrepresentation of specific GO terms, with lower p-values suggesting higher significance.

## Statistics and reproducibility

All genomic data utilized in this study, including the species-specific datasets, were obtained from publicly available sources. Statistical analyses, such as the Student's t-test, Wilcoxon test, correlation test, and linear regression analysis, were performed using GraphPad Prism software. The samples used in the statistical analyses corresponded to the genomic data, PAIs, or specific genes under investigation.

## Acknowledgements

The sincere appreciation extends to Dr. Sung Ho Yoon and his colleagues for their dedicated efforts in identifying PAIs and establishing the Pathogenicity Island Database for public analysis. Their commitment to advancing the field of pathogen genomics has greatly facilitated this research. This study would like to thank Dr. Jingjing Li (Zhejiang University) and Dr. Mingyu Zhou (Sun Yat-Sen University) for their insightful suggestions and constructive comments regarding the exploration of G4 structures in genomes. Their expertise and guidance have significantly enriched the understanding of the potential roles and implications of G4 structures in the context of PAIs.

## Additional information

### Funding
No external funding was received for this work.

### Author contributions
Bo Lyu, Conceptualization, Data curation, Software, Formal analysis, Writing – original draft; Qisheng Song, Conceptualization, Supervision, Writing – review and editing

### Author ORCIDs
Bo Lyu ⓘ https://orcid.org/0000-0002-2744-730X
Qisheng Song ⓘ https://orcid.org/0000-0001-9682-1775

Reviewer #1 (Public Review): https://doi.org/10.7554/eLife.91985.3.sa1
Reviewer #2 (Public Review): https://doi.org/10.7554/eLife.91985.3.sa2
Author Response https://doi.org/10.7554/eLife.91985.3.sa3

## Additional files

### Supplementary files
• Supplementary file 1. The genomic features of pathogens and pathogenic islands, as well as the putative functions of G4s across two *E. coli* strains. (**a**) List of 89 bacterial species or strains within the same species and the number of G4s within their genomic features. (**b**) List of pathogenicity islands (PAIs) reported to be present in genome sequences and the number of G4s within PAIs. (**c**) Functional annotations for *E. coli* strain 1. (**d**) Functional annotations for *E. coli* strain 2.
• MDAR checklist

### Data availability
The original reported PAIs datasets analyzed in this study are available from the publication *Yoon et al., 2015*. Additionally, *Supplementary file 1* provides further PAIs data analyzed in the study.

The following previously published datasets were used:

| Author(s) | Year | Dataset title | Dataset URL | Database and Identifier |
|---|---|---|---|---|
| Saenz HL | 2007 | Intracellular pathogen isolated from wild rats | https://www.ncbi.nlm.nih.gov/bioproject/PRJNA28109/ | NCBI BioProject, PRJNA28109 |
| Gartemann KH | 2008 | Phytopathogen that causes bacterial wilt and canker of tomato | https://www.ncbi.nlm.nih.gov/bioproject/PRJNA19643/ | NCBI BioProject, PRJNA19643 |
| University of Helsinki | 2021 | Clostridium perfringens isolates and their heat resistance | https://www.ncbi.nlm.nih.gov/bioproject/PRJNA707150/ | NCBI BioProject, PRJNA707150 |
| Bielefeld University | 2012 | Corynebacterium diphtheriae 241 genome sequencing | https://www.ncbi.nlm.nih.gov/bioproject/PRJNA42407/ | NCBI BioProject, PRJNA42407 |
| Bielefeld University | 2012 | Corynebacterium diphtheriae C7 (beta) genome sequencing | https://www.ncbi.nlm.nih.gov/bioproject/PRJNA42401/ | NCBI BioProject, PRJNA42401 |
| Bielefeld University | 2012 | Corynebacterium diphtheriae CDCE 8392 genome sequencing | https://www.ncbi.nlm.nih.gov/bioproject/PRJNA42405/ | NCBI BioProject, PRJNA42405 |
| Bielefeld University | 2012 | Corynebacterium diphtheriae HC03 genome sequencing | https://www.ncbi.nlm.nih.gov/bioproject/PRJNA42415/ | NCBI BioProject, PRJNA42415 |
| Bielefeld University | 2012 | Corynebacterium diphtheriae INCA 402 genome sequencing | https://www.ncbi.nlm.nih.gov/bioproject/PRJNA42419/ | NCBI BioProject, PRJNA42419 |
| Sanger Institute | 2003 | Causative agent of diphtheria | https://www.ncbi.nlm.nih.gov/bioproject/PRJNA87/ | NCBI BioProject, PRJNA87 |
| Bielefeld University | 2012 | Corynebacterium diphtheriae PW8 genome sequencing | https://www.ncbi.nlm.nih.gov/bioproject/PRJNA42403/ | NCBI BioProject, PRJNA42403 |
| Bielefeld University | 2012 | Corynebacterium pseudotuberculosis 1002 genome sequencing | https://www.ncbi.nlm.nih.gov/bioproject/PRJNA40687/ | NCBI BioProject, PRJNA40687 |
| University Federal of Minas Gerais | 2020 | Corynebacterium pseudotuberculosis strain C231, whole genome sequencing | https://www.ncbi.nlm.nih.gov/bioproject/PRJNA40875/ | NCBI BioProject, PRJNA40875 |
| Bielefeld University | 2019 | Corynebacterium pseudotuberculosis FRC41 genome sequencing project | https://www.ncbi.nlm.nih.gov/bioproject/PRJNA48979/ | NCBI BioProject, PRJNA48979 |
| Rede Paraense de Genômica e Proteômica | 2019 | Corynebacterium pseudotuberculosis I19 genome sequencing | https://www.ncbi.nlm.nih.gov/bioproject/PRJNA52845/ | NCBI BioProject, PRJNA52845 |
| The Enterobacter sakazakii Genome Sequencing Project | 2007 | Isolated from dried infant formula and causes infant septicemia | https://www.ncbi.nlm.nih.gov/bioproject/PRJNA12720/ | NCBI BioProject, PRJNA12720 |
| TIGR | 2003 | Opportunistic pathogen that transfers vancomycin resistance to other bacteria | https://www.ncbi.nlm.nih.gov/bioproject/PRJNA70/ | NCBI BioProject, PRJNA70 |
| Baylor College of Medicine | 2012 | reference genome for the Human Microbiome Project | https://www.ncbi.nlm.nih.gov/bioproject/PRJNA30627/ | NCBI BioProject, PRJNA30627 |

*Continued*

| Author(s) | Year | Dataset title | Dataset URL | Database and Identifier |
|---|---|---|---|---|
| University Iowa State | 2006 | Avian pathogenic strain | https://www.ncbi.nlm.nih.gov/bioproject/PRJNA16718/ | NCBI BioProject, PRJNA16718 |
| Genentech | 2020 | *Escherichia coli* CFT073 isolate:199310 Genome sequencing | https://www.ncbi.nlm.nih.gov/bioproject/PRJNA624646/ | NCBI BioProject, PRJNA624646 |
| University of Tokyo | 2009 | Enterohemorrhagic strain | https://www.ncbi.nlm.nih.gov/bioproject/PRJDA32509/ | NCBI BioProject, PRJDA32509 |
| University of Tokyo | 2009 | This strain will be used for comparative genome analysis | https://www.ncbi.nlm.nih.gov/bioproject/PRJDA32511/ | NCBI BioProject, PRJDA32511 |
| University of Tokyo | 2009 | This strain will be used for comparative genome analysis | https://www.ncbi.nlm.nih.gov/bioproject/PRJDA32513/ | NCBI BioProject, PRJDA32513 |
| University of California San Diego | 2014 | *Escherichia coli* O157:H7 str. EDL933 Genome sequencing | https://www.ncbi.nlm.nih.gov/bioproject/PRJNA253471/ | NCBI BioProject, PRJNA253471 |
| GIRC | 2018 | Enterohemorrhagic *Escherichia coli* | https://www.ncbi.nlm.nih.gov/bioproject/PRJNA226/ | NCBI BioProject, PRJNA226 |
| Genoscope | 2008 | Urinary tract infection isolate | https://www.ncbi.nlm.nih.gov/bioproject/PRJNA33415/ | NCBI BioProject, PRJNA33415 |
| DOE Joint Genome Institute | 2008 | Causative agent of tularemia | https://www.ncbi.nlm.nih.gov/bioproject/PRJNA19571/ | NCBI BioProject, PRJNA19571 |
| Los Alamos National Laboratory | 2015 | *Francisella tularensis* tularensis Schu_S4 Genome sequencing | https://www.ncbi.nlm.nih.gov/bioproject/PRJNA239340/ | NCBI BioProject, PRJNA239340 |
| BioHealthBase | 2007 | Causative agent of tularemia | https://www.ncbi.nlm.nih.gov/bioproject/PRJNA18459/ | NCBI BioProject, PRJNA18459 |
| Wuerzburg Univ | 2003 | Causes hepatitis, typhlitis, hepatocellular tumors, and gastric bowel disease | https://www.ncbi.nlm.nih.gov/bioproject/PRJNA185/ | NCBI BioProject, PRJNA185 |
| RIPCM | 2012 | Helicobacter pylori 26695 Genome sequencing | https://www.ncbi.nlm.nih.gov/bioproject/PRJNA175543/ | NCBI BioProject, PRJNA175543 |
| Bielefeld University | 2010 | Helicobacter pylori B8 genome sequencing project | https://www.ncbi.nlm.nih.gov/bioproject/PRJEA41831/ | NCBI BioProject, PRJEA41831 |
| The University of Tokyo | 2011 | Helicobacter pylori F16 genome sequencing project | https://www.ncbi.nlm.nih.gov/bioproject/PRJDA50589/ | NCBI BioProject, PRJDA50589 |
| The University of Tokyo | 2011 | Helicobacter pylori F30 genome sequencing project | https://www.ncbi.nlm.nih.gov/bioproject/PRJDA50591/ | NCBI BioProject, PRJDA50591 |
| The University of Tokyo | 2011 | Helicobacter pylori F32 genome sequencing project | https://www.ncbi.nlm.nih.gov/bioproject/PRJDA50593/ | NCBI BioProject, PRJDA50593 |
| The University of Tokyo | 2011 | Helicobacter pylori F57 genome sequencing project | https://www.ncbi.nlm.nih.gov/bioproject/PRJDA50595/ | NCBI BioProject, PRJDA50595 |

*Continued*

| Author(s) | Year | Dataset title | Dataset URL | Database and Identifier |
|---|---|---|---|---|
| Icahn School of Medicine at Mount Sinai | 2015 | Multi-strain, long-read bacterial genome sequencing | https://www.ncbi.nlm. nih.gov/bioproject/ PRJNA281410/ | NCBI BioProject, PRJNA281410 |
| Max von Pettenkofer-Institut für Hygiene und Medizinische Mikrobiologie, Ludwig-Maximilians-Universität München | 2008 | Clinical isolate | https://www.ncbi.nlm. nih.gov/search/all/? term=PRJNA32291 | NCBI BioProject, PRJNA32291 |
| Wuhan Institute of Virology, Chinese Academy of Sciences | 2008 | Mosquito larvae pathogen | https://www.ncbi.nlm. nih.gov/bioproject/ PRJNA19619/ | NCBI BioProject, PRJNA19619 |
| Microbial Genome Center of ChMPH | 2007 | Unknown strain | https://www.ncbi.nlm. nih.gov/bioproject/ PRJNA16393/ | NCBI BioProject, PRJNA16393 |
| TIGR | 2005 | Causes meningitis and septicemia | https://www.ncbi.nlm. nih.gov/bioproject/ PRJNA251/ | NCBI BioProject, PRJNA251 |
| IREC | 2010 | Rhodococcus equi strain 103S whole genome sequencing project | https://www.ncbi.nlm. nih.gov/bioproject/ PRJEA41335/ | NCBI BioProject, PRJEA41335 |
| Sanger Institute | 2008 | An opportunistic pathogen in normal gut flora | https://www.ncbi.nlm. nih.gov/bioproject/ PRJNA12624/ | NCBI BioProject, PRJNA12624 |
| TIGR | 2003 | Causes plant rot | https://www.ncbi.nlm. nih.gov/bioproject/ PRJNA359/ | NCBI BioProject, PRJNA359 |
| Chang Gung Genomic Medical Center, Chang Gung Memorial Hospital | 2005 | Extremely invasive *Salmonella* that causes severe disease in pigs and humans | https://www.ncbi.nlm. nih.gov/bioproject/ PRJNA9618/ | NCBI BioProject, PRJNA9618 |
| Sanger Institute | 2003 | Human-specific *Salmonella* that causes Typhoid fever | https://www.ncbi.nlm. nih.gov/bioproject/ PRJNA236/ | NCBI BioProject, PRJNA236 |
| Wisconsin Univ | 2003 | Human-specific *Salmonella* that causes Typhoid fever | https://www.ncbi.nlm. nih.gov/bioproject/ PRJNA371/ | NCBI BioProject, PRJNA371 |
| Washington University Genome Sequencing Center | 2016 | Major laboratory strain of *Salmonella* typhimurium | https://www.ncbi.nlm. nih.gov/bioproject/ PRJNA241/ | NCBI BioProject, PRJNA241 |
| Microbial Genome Center of ChMPH | 2011 | Human-specific pathogen that causes endemic dysentery | https://www.ncbi.nlm. nih.gov/bioproject/ PRJNA310/ | NCBI BioProject, PRJNA310 |
| Wisconsin Univ | 2003 | Human-specific pathogen that causes endemic dysentery | https://www.ncbi.nlm. nih.gov/bioproject/ PRJNA408/ | NCBI BioProject, PRJNA408 |
| Minnesota Univ | 2005 | Associated with mastitis in cattle | https://www.ncbi.nlm. nih.gov/bioproject/ PRJNA63/ | NCBI BioProject, PRJNA63 |
| IntegratedGenomics | 2013 | *Staphylococcus aureus* subsp. aureus CN1 Genome sequencing | https://www.ncbi.nlm. nih.gov/bioproject/ PRJNA162343/ | NCBI BioProject, PRJNA162343 |
| TIGR | 2005 | Methicillin resistant strain | https://www.ncbi.nlm. nih.gov/bioproject/ PRJNA238/ | NCBI BioProject, PRJNA238 |

*Continued on next page*

*Continued*

| Author(s) | Year | Dataset title | Dataset URL | Database and Identifier |
|---|---|---|---|---|
| University of Edinburgh | 2009 | *Staphylococcus aureus ED98 genome sequencing* | https://www.ncbi.nlm.nih.gov/bioproject/PRJNA39547/ | NCBI BioProject, PRJNA39547 |
| University of Edinburgh | 2010 | *Staphylococcus aureus subsp. aureus ED133 genome sequencing project* | https://www.ncbi.nlm.nih.gov/bioproject/PRJNA41277/ | NCBI BioProject, PRJNA41277 |
| Sanger Institute | 2004 | Methicillin resistant strain from the UK | https://www.ncbi.nlm.nih.gov/bioproject/PRJNA265/ | NCBI BioProject, PRJNA265 |
| Sanger Institute | 2004 | Methicillin sensitive strain from the UK | https://www.ncbi.nlm.nih.gov/bioproject/PRJNA266/ | NCBI BioProject, PRJNA266 |
| Univ Juntendo | 2004 | Methicillin and vancomycin resistant strain | https://www.ncbi.nlm.nih.gov/bioproject/PRJNA263/ | NCBI BioProject, PRJNA263 |
| NITE | 2004 | Methicillin resistant strain | https://www.ncbi.nlm.nih.gov/bioproject/PRJNA306/ | NCBI BioProject, PRJNA306 |
| Univ Juntendo | 2004 | Methicillin resistant strain | https://www.ncbi.nlm.nih.gov/bioproject/PRJNA264/ | NCBI BioProject, PRJNA264 |
| University Medical Centre Utrecht | 2010 | *Staphylococcus aureus subsp. aureus ST398* | https://www.ncbi.nlm.nih.gov/bioproject/PRJEA29427/ | NCBI BioProject, PRJEA29427 |
| Juntendo University School of Medicine, Department of Bacteriology | 2007 | An opportunistic pathogen in humans and animals | https://www.ncbi.nlm.nih.gov/bioproject/PRJDA18801/ | NCBI BioProject, PRJDA18801 |
| University of California, San Francisco | 2006 | A methicillin resistant strain of *Staphylococcus aureus* | https://www.ncbi.nlm.nih.gov/bioproject/PRJNA16313/ | NCBI BioProject, PRJNA16313 |
| Chinese National HGC Shanghai | 2002 | Used for detection of residual antibiotics in food products | https://www.ncbi.nlm.nih.gov/bioproject/PRJNA279/ | NCBI BioProject, PRJNA279 |
| TIGR | 2005 | Pathogenic clinical isolate that causes toxic-shock syndrome and staphylococcal scarlet fever | https://www.ncbi.nlm.nih.gov/bioproject/PRJNA64/ | NCBI BioProject, PRJNA64 |
| The Wellcome Trust Sanger Institute | 2009 | Causes strangles disease | https://www.ncbi.nlm.nih.gov/bioproject/PRJEA30765/ | NCBI BioProject, PRJEA30765 |
| Herz- und Diabeteszentrum Nordrhein-Westfalen Universitätsklinik der Ruhr-Universität Bochum | 2011 | Streptococcus gallolyticus subsp. galloyticus ATCC BAA-2069 genome sequencing | https://www.ncbi.nlm.nih.gov/bioproject/PRJEA63179/ | NCBI BioProject, PRJEA63179 |
| Department ofMicrobiology University of Kaiserslautern | 2010 | Clinical isolate | https://www.ncbi.nlm.nih.gov/bioproject/PRJNA16302/ | NCBI BioProject, PRJNA16302 |
| UniversityChang Gung | 2012 | Streptococcus parasanguinis FW213 Genome sequencing | https://www.ncbi.nlm.nih.gov/bioproject/PRJNA76769/ | NCBI BioProject, PRJNA76769 |

*Continued*

| Author(s) | Year | Dataset title | Dataset URL | Database and Identifier |
|---|---|---|---|---|
| Wellcome Trust Sanger Institute | 2009 | multidrug resistant strain | https://www.ncbi.nlm.nih.gov/bioproject/PRJEA31233/ | NCBI BioProject, PRJEA31233 |
| Institute Broad | 2012 | Genome sequencing with short reads | https://www.ncbi.nlm.nih.gov/bioproject/PRJNA76613/ | NCBI BioProject, PRJNA76613 |
| Lab of Human Bacterial Pathogenesis | 2005 | Causative agent of a wide range of human and animal infections | https://www.ncbi.nlm.nih.gov/bioproject/PRJNA13888/ | NCBI BioProject, PRJNA13888 |
| Lab of Human Bacterial Pathogenesis | 2005 | Causative agent of a wide range of human and animal infections | https://www.ncbi.nlm.nih.gov/bioproject/PRJNA13887/ | NCBI BioProject, PRJNA13887 |
| Beijing Institute of Genomics, Chinese Academy of Sciences | 2007 | Causes disease in pigs and occasionally humans | https://www.ncbi.nlm.nih.gov/bioproject/PRJNA17153/ | NCBI BioProject, PRJNA17153 |
| Beijing Institute of Genomics, Chinese Academy of Sciences | 2007 | Causes disease in pigs and occasionally humans | https://www.ncbi.nlm.nih.gov/bioproject/PRJNA17155/ | NCBI BioProject, PRJNA17155 |
| Wellcome Sanger Institute | 2019 | Updated VC N16961 reference genome | https://www.ncbi.nlm.nih.gov/bioproject/PRJEB22249/ | NCBI BioProject, PRJEB22249 |
| Centers for Disease Control and Prevention | 2011 | *Vibrio cholerae* O1 str. 2010EL-1786 genome sequencing project | https://www.ncbi.nlm.nih.gov/bioproject/PRJNA59943/ | NCBI BioProject, PRJNA59943 |
| University of Oslo | 2019 | *Vibrio cholerae* O395 isolate:TCP2 Genome sequencing | https://www.ncbi.nlm.nih.gov/bioproject/PRJNA586749/ | NCBI BioProject, PRJNA586749 |
| Sao Paulostate (Brazil) Consortium | 2003 | Plant-specific pathogen that causes citrus canker | https://www.ncbi.nlm.nih.gov/bioproject/PRJNA297/ | NCBI BioProject, PRJNA297 |
| Chinese National HGC Shanghai | 2005 | Causes black rot and citrus canker | https://www.ncbi.nlm.nih.gov/bioproject/PRJNA15/ | NCBI BioProject, PRJNA15 |
| Sao Paulostate (Brazil) Consortium | 2002 | Plant-specific pathogen that causes black rot | https://www.ncbi.nlm.nih.gov/bioproject/PRJNA296/ | NCBI BioProject, PRJNA296 |
| GenomicsBacterial, LaboratoryEvolution and CSIR-Institute of Microbial Technology, Sector 39-A, Chandigarh, India | 2017 | Xanthomonas campestris pv. vitistrifoliae strain:LMG940 Genome sequencing and assembly | https://www.ncbi.nlm.nih.gov/bioproject/PRJNA298596/ | NCBI BioProject, PRJNA298596 |
| NIAB, Rural Development Administration | 2005 | Causes rice bacterial blight disease | https://www.ncbi.nlm.nih.gov/bioproject/PRJNA12931/ | NCBI BioProject, PRJNA12931 |
| Sanger Institute | 2007 | Food and waterborn pathogen that causes gastroenteritis | https://www.ncbi.nlm.nih.gov/bioproject/PRJNA190/ | NCBI BioProject, PRJNA190 |
| Academy of Military Medical Sciences, The Institute of Microbiology and Epidemiology, China | 2004 | Extremely virulent organism that causes plague | https://www.ncbi.nlm.nih.gov/bioproject/PRJNA10638/ | NCBI BioProject, PRJNA10638 |

*Continued on next page*

*Continued*

| Author(s) | Year | Dataset title | Dataset URL | Database and Identifier |
|---|---|---|---|---|
| Sanger Institute | 2003 | Extremely virulent organism that causes plague | https://www.ncbi.nlm.nih.gov/bioproject/PRJNA34/ | NCBI BioProject, PRJNA34 |
| J. Craig Venter Institute | 2009 | Yersinia pestis KIM D27 genome sequencing project | https://www.ncbi.nlm.nih.gov/bioproject/PRJNA41469/ | NCBI BioProject, PRJNA41469 |
| TIGR | 2007 | Serotype 1b strain isolated from a patient in Russia | https://www.ncbi.nlm.nih.gov/bioproject/PRJNA16070/ | NCBI BioProject, PRJNA16070 |
| Los Alamos National Laboratory | 2015 | Yersinia pseudotuberculosis IP 32953 Genome sequencing | https://www.ncbi.nlm.nih.gov/bioproject/PRJNA239344/ | NCBI BioProject, PRJNA239344 |
| The Wellcome Trust Sanger Institute | 2011 | *Staphylococcus aureus* subsp. aureus MSHR1132 genome sequencing project | https://www.ncbi.nlm.nih.gov/bioproject/PRJEA62885/ | NCBI BioProject, PRJEA62885 |

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
