## [Editor Report · eLife assessment]

This **fundamental** study explores the relationship between guanine-quadruplex structures and pathogenicity islands in 89 bacterial strains representing a range of pathogens. Guanine-quadruplex structures were found to be non-randomly distributed within pathogenicity islands and conserved within the same strains. These **compelling** findings shed light on the molecular mechanisms of Guanine-quadruplex structure-pathogenicity island interactions and will be of interest to all microbiologists.

---

## [Referee Report · Reviewer #1 (Public Review)]

Summary:

This study explores the relationship between guanine-quadruplex (G4) structures and pathogenicity islands (PAIs) in 89 pathogenic strains.

Strengths:

The findings of this study hold significant implications for our understanding of bacterial pathogenicity and the role of guanine-quadruplex (G4) structures:

Molecular Mechanisms of Pathogenicity: The study highlights that G4 structures are not randomly distributed within pathogenicity islands (PAIs), suggesting a potential role in regulating pathogenicity. This insight into the uneven distribution of G4s within PAIs provides a basis for further research into the molecular mechanisms underlying bacterial pathogenicity.

Conservation of G4 Structures: The consistent conservation of G4 structures within the same pathogenic strains suggests that these structures might play a vital and possibly conserved role in the pathogenicity of these bacteria. This finding opens doors for exploring how G4s influence virulence across different pathogens.

Unique Nature of PAIs: The differences in GC content between PAIs and the core genome underscore the unique nature of PAIs. This distinction suggests that factors such as DNA topology and G4 structures might contribute to the specialized functions and characteristics of PAIs, which are often associated with virulence genes.

Regulatory Role of G4s: The identification of high-confidence G4 structures within regulatory regions of *Escherichia coli* implies that these structures could influence the efficiency or specificity of DNA integration events within PAIs. This finding provides a potential mechanism by which G4s can impact the pathogenicity of bacteria.

Weaknesses:

None

Overall, the study provides fundamental insights into the pathogenicity island and conservation of G4 motifs.

---

## [Referee Report · Reviewer #2 (Public Review)]

Summary: In the mauscript entitled "The Intricate Relationship of G-Quadruplexes and Pathogenicity Islands: A Window into Bacterial Pathogenicity" Bo Lyu explored the interactions between guanine-quadruplex (G4) structures and pathogenicity islands (PAIs) in 89 bacterial genomes through rigorous computational approach. This paper handles an intriguing and complex topic in the field pathogenomics, it has the potential to contribute significantly to the understanding of G4-PAI interactions and bacterial pathogenicity.

Strengths: Chosen research area and summarizing the results through neat illustrations

Weaknesses: I did not find any significant ones.

---

## [Author Response]

The following is the authors’ response to the original reviews.

**Reviewer #1 (Public Review):**
Summary:This study explores the relationship between guanine-quadruplex (G4) structures and pathogenicity islands (PAIs) in 89 pathogenic strains. G4 structures were found to be non-randomly distributed within PAIs and conserved within the same strains. Positive correlations were observed between G4s and GC content across various genomic features, suggesting a link between G4 structures and GC-rich regions. Differences in GC content between PAIs and the core genome underscored the unique nature of PAIs. High-confidence G4 structures in *Escherichia coli*'s regulatory regions were identified, influencing DNA integration within PAIs. These findings shed light on the molecular mechanisms of G4-PAI interactions, enhancing our understanding of bacterial pathogenicity and G4 structures in infectious diseases.Strengths:The findings of this study hold significant implications for our understanding of bacterial pathogenicity and the role of guanine-quadruplex (G4) structures.Molecular Mechanisms of Pathogenicity: The study highlights that G4 structures are not randomly distributed within pathogenicity islands (PAIs), suggesting a potential role in regulating pathogenicity. This insight into the uneven distribution of G4s within PAIs provides a basis for further research into the molecular mechanisms underlying bacterial pathogenicity.Conservation of G4 Structures: The consistent conservation of G4 structures within the same pathogenic strains suggests that these structures might play a vital and possibly conserved role in the pathogenicity of these bacteria. This finding opens doors for exploring how G4s influence virulence across different pathogens.Unique Nature of PAIs: The differences in GC content between PAIs and the core genome underscore the unique nature of PAIs. This distinction suggests that factors such as DNA topology and G4 structures might contribute to the specialized functions and characteristics of PAIs, which are often associated with virulence genes.Regulatory Role of G4s: The identification of high-confidence G4 structures within regulatory regions of *Escherichia coli* implies that these structures could influence the efficiency or specificity of DNA integration events within PAIs. This finding provides a potential mechanism by which G4s can impact the pathogenicity of bacteria.Weaknesses:No weaknesses were identified by this reviewer.Overall, the study provides fundamental insights into the pathogenicity island and conservation of G4 motifs.

Thank you for your thorough review of our manuscript exploring the relationship between G4 structures and PAIs in 89 pathogenic strains. We appreciate your recognition of the strengths of our study and its potential implications for understanding bacterial pathogenicity. We are pleased that you highlighted the significance of our findings in revealing the non-random distribution and conservation of G4 structures within PAIs across various pathogenic strains.

Your insightful comments about the molecular mechanisms of pathogenicity, the conservation of G4 structures, the unique nature of PAIs, and the regulatory role of G4s within *Escherichia coli* are invaluable. We are encouraged by your positive evaluation of these aspects, which underscores the potential impact of our work on advancing the understanding of bacterial pathogenicity.

**Reviewer #2 (Public Review):**
Summary:In the manuscript entitled "The Intricate Relationship of G-Quadruplexes and Pathogenicity Islands: A Window into Bacterial Pathogenicity" Bo Lyu explored the interactions between guanine-quadruplex (G4) structures and pathogenicity islands (PAIs) in 89 bacterial genomes through a rigorous computational approach. This paper handles an intriguing and complex topic in the field of pathogenomics. It has the potential to contribute significantly to the understanding of G4-PAI interactions and bacterial pathogenicity.Strengths:The chosen research area.The summarizing of the results through neat illustrations.Weaknesses:This reviewer did not find any significant weaknesses.

Thank you for your positive and encouraging feedback on our manuscript. We appreciate your specific mention of the strengths, particularly highlighting the chosen research area and the effectiveness of our illustrations in summarizing the results. Your acknowledgment of these aspects is motivating, and we are pleased that the content and presentation resonated well with you.

**Reviewer #3 (Public Review):**
The main problem with the work is that the results are only descriptive and do not allow any inferences or conclusions about the importance of the function of G4 structures. The discussion and conclusions are poor. The results are preliminary and in order to try to make the analysis more interesting, it should be further extended and the data must be explored in a much greater depth.

Thank you for your constructive feedback on our manuscript, and appreciate the time and effort you dedicated to evaluating our work. We acknowledge your concern regarding the descriptive nature of the results and the limitations in making inferences about the importance of G4 structures. To address this, we plan to enhance the depth of our analysis and provide more insightful interpretations in the discussion and conclusion sections. It's important to note that this study is intentionally a short report, emphasizing data mining findings rather than laboratory results. We understand the value of in-depth investigations and concur that our work lays the groundwork for more extensive studies in this area, aiming to provide a real-world scenario. We are committed to addressing your comments and refining our manuscript to contribute meaningfully to this field. Your insights are invaluable, and we look forward to presenting an improved version of our study.

**Reviewer #2 (Recommendations For The Authors):**
The authors could try a higher G-quadruplex score of 1.4 or higher values to substantiate their findings or pick up the bacterial genomes that relied on G4s for their pathogenecity.

We acknowledge your recommendation to explore a higher G-quadruplex score, and we would like to assure you that we have already conducted analyses using thresholds of 1.4 and 1.6. The findings consistently support the observations presented in the manuscript. We have updated the text to reflect this additional analysis, and the results are included in the revised version of the manuscript (Figure S1).

**Reviewer #3 (Recommendations For The Authors):**
Minor pointsIntroductionQ1. The introduction is shallow. The concept and the importance of PAIs is vague. Why should these genes be different from other genes?

A1: Thank you for your valuable feedback and we have incorporated additional content to provide a more comprehensive understanding of PAIs and their distinctiveness from other genes in the Introduction section.

Changes: Lines 44-49 “G4 structures are ...innovative technologies.” were added.

Lines 51-55 “PAIs are distinct...such as plasmids.” were added.

Lines 60-66 “PAIs typically contain...recipient genome” were added.

Lines 77-80 “Growing evidence has...CpG islands, and PAIs” were added.

Material and MethodsQ2. It is not clear if the author used the TBTools or the G4Hunter software G4 structures. It would be interesting to include references to published articles that used this software.

A2: Thank you! Corrected and added more references that used TBTools to extract sequences and G4Hunter to identify G4 structures.

Q3. The statistical significance must not be based only on p-values. P-values are influenced by sample sizes. I strongly recommend the use of other parameters such as confidence interval and ROC analysis.

A3: Thank you! We have incorporated confidence intervals and ROC analysis to complement p-values, enhancing the robustness of our statistical analysis.

Changes: Lines 265-267 “The correlation's significance... sensitivity and specificity.” were added.

Results and discussionQ4. The stability of G4 structures seems to be important for its function (doi:10.1111/febs.15065). Therefore it would be interesting if the analysis were carried out separating the G4 according to stability.

A4: Thank you for highlighting the importance of G4 structure stability for its function and suggesting an analysis based on stability. We have carefully reviewed the referenced paper (doi:10.1111/febs.15065) and note that their study focused on the stability analysis of individual G4s. In our current study, we identified a large number of G4s, and while stability analysis for each G4 is indeed an interesting avenue, it goes beyond the scope of this particular investigation. However, we agree that exploring the relationship between G4 stability and function is a valuable topic. We plan to delve deeper into this aspect in future work, as discussed in our response to your previous comment.

Changes: Lines 217-221 “Lastly, the stability of G4...molecular engineering.” were added.

Q5. The quality of the figures is poor. Is not possible to read the correlation and p-values from Figure 2.

A5: The revised figure is now submitted with enhanced clarity to ensure that correlation and p-values can be easily discerned.

Q6. The analysis of promoter regions should be performed taking into account the distance between the G4 and the beginning of the gene.

A6: Thank you and we have elaborated more in the revision.

Changes: Lines 198-106 “Additionally, considering the distance...of G4 structures in promoters.” were added.

Q7. The topic "Putative origin, transfer mechanisms, and functions of G4s in PAIs". The comments made on this topic are purely speculative and not backed up by data or any type of experimental analysis.

A7: We appreciate the feedback and have revised the title to emphasize the focus on the functions of G4s in PAIs. We acknowledge that the content related to the putative origin and transfer mechanisms of G4s in PAIs is purely descriptive and speculative, we have made the adjustment to relocate this information to the discussion section for a more appropriate treatment.

Q8. The supplemental material is hard to follow. The meaning of each column should be better explained. Why was the data divided into 10 parts?

A8: Following your suggestion, we have revised the tables for better clarity. To address concerns about the division into 10 parts, we have decided to remove this data from the tables as it was deemed unnecessary for presentation.

Q9. Why was the data of *E. coli* strains 1 and 2 shown in Tables S3 and S4 and the other bacterial strains were not?

A9: We appreciate your inquiry. The data of *E. coli* strains 1 and 2 were specifically highlighted in Tables S3 and S4 as illustrative examples to demonstrate the putative functions of G4s in PAIs within the scope of our study. Given the extensive nature of function annotation analyses across various pathogenic strains, presenting additional tables for each strain would have resulted in an impractical volume of supplementary material.

Q10. The Results and Discussion should be separated.

A10: Thank you! Corrected as suggested.